# Elimination of Vitamin D Signaling Causes Increased Mortality in a Model of Overactivation of the Insulin Receptor: Role of Lipid Metabolism

**DOI:** 10.3390/nu14071516

**Published:** 2022-04-05

**Authors:** Maria Crespo-Masip, Aurora Perez-Gomez, Alicia Garcia-Carrasco, Ramiro Jover, Carla Guzmán, Xavier Dolcet, Mercé Ibarz, Cristina Martínez, Àuria Eritja, Juan Miguel Diaz-Tocados, José Manuel Valdivielso

**Affiliations:** 1Vascular and Renal Translational Research Group, Biomedical Research Institute of Lleida, Dr. Pifarré Foundation (IRBLleida), 25198 Lleida, Spain; mcmasip@hotmail.com (M.C.-M.); auroraperezgomez@gmail.com (A.P.-G.); agarcia@irblleida.cat (A.G.-C.); cristina.martinez@vhir.org (C.M.); aeritja@irblleida.cat (À.E.); 2Medicine Department, University of Lleida, 25198 Lleida, Spain; 3Experimental Hepatology Unit, IIS Hospital La Fe, 46026 Valencia, Spain; ramiro.jover@uv.es (R.J.); carla_teclab@yahoo.es (C.G.); 4Research Network Center for Liver and Digestive Diseases (CIBERehd), 28029 Madrid, Spain; 5Biochemistry and Molecular Biology Department, University of Valencia, 46100 Valencia, Spain; 6Oncological Pathology Research Group, IRBLleida, 25198 Lleida, Spain; xavi.dolcet@udl.cat; 7Indicators and Specifications of the Quality in the Clinical Laboratory Group, IRBLleida, 25198 Lleida, Spain; mibarz.lleida.ics@gencat.cat

**Keywords:** hypoglycemia, diabetes, insulin overdose, fatty acids, lipolysis

## Abstract

Vitamin D (VD) deficiency has been associated with cancer and diabetes. Insulin signaling through the insulin receptor (IR) stimulates cellular responses by activating the PI3K/AKT pathway. PTEN is a tumor suppressor and a negative regulator of the pathway. Its absence enhances insulin signaling leading to hypoglycemia, a dangerous complication found after insulin overdose. We analyzed the effect of VD signaling in a model of overactivation of the IR. We generated inducible double KO (DKO) mice for the VD receptor (VDR) and PTEN. DKO mice showed severe hypoglycemia, lower total cholesterol and increased mortality. No macroscopic tumors were detected. Analysis of the glucose metabolism did not show clear differences that would explain the increased mortality. Glucose supplementation, either systemically or directly into the brain, did not enhance DKO survival. Lipidic liver metabolism was altered as there was a delay in the activation of genes related to β-oxidation and a decrease in lipogenesis in DKO mice. High-fat diet administration in DKO significantly improved its life span. Lack of vitamin D signaling increases mortality in a model of overactivation of the IR by impairing lipid metabolism. Clinically, these results reveal the importance of adequate Vitamin D levels in T1D patients.

## 1. Introduction

Type 1 diabetes mellitus (T1D) develops as a consequence of pancreatic beta-cell destruction and it is characterized by insulin deficiency, a tendency to ketosis and dependence on exogenous insulin to sustain life. Glycemic control in type 1 diabetes is of paramount importance, as it has been demonstrated to be clearly associated with long term complications. Thus, there is unquestionable evidence of a very close relationship between hemoglobin A1c (HbA1c) concentrations, maintained over the long term, and the onset or progression of microvascular and macrovascular complications [1,2]. Therefore, insulin administration is routinely used in T1D patients. There has been a dramatic increase in the types of insulin available (rapid, intermediate and long acting) to help with the management of glucose levels. However, this has also led to an increase in dosing errors, which are considered to be some of the most dangerous medication errors that can occur, occasionally leading to profound hypoglycemia and death [3]. Indeed, conventional therapeutic management with insulin leads to an average one or two episodes of symptomatic hypoglycemia weekly and at least one episode of severe, temporarily disabling hypoglycemia per year [1,4].

Insulin signaling is mediated by the insulin receptor, which belongs to the superfamily of receptor tyrosine kinases (RTKs) [5]. The pathway emanating from the insulin receptor responsible for most metabolic effects of insulin is the phosphatidylinositol 3-kinase (PI3K) pathway [6]. After pathway initiation, PI3K generates the second messenger (PIP3), which subsequently activates the serine/threonine protein kinase B (PKB/AKT). This leads to serine and/or threonine phosphorylation of a range of downstream substrates, which are often kinase/phosphatases or other signaling molecules [7]. A critical upstream member of the cascade is the phosphatase and tensin homolog (PTEN), a dual-specificity lipid and protein phosphatase that efficiently dephosphorylates the 3′-group of PIP3, and therefore, terminates propagation of the signal to AKT and other PIP3-effector targets [8]. The presence and activation of PTEN at the cytoplasmic membrane are crucial to guarantee controlled transduction of the PI3K signal, which is then transmitted to the cell. Thus, PTEN deficiency in humans enhances insulin signaling [9], a similar situation to that seen in individuals with repeated episodes of insulin-induced hypoglycemia.

Vitamin D, signaling through the vitamin D receptor (VDR), has well documented effects on calcium homeostasis and bone metabolism but several studies suggest a much broader role for this secosteroid in human health. Indeed, vitamin D deficiency has been involved in many other conditions related to the immune, cardiovascular, endocrine, respiratory and even reproductive systems [10,11,12,13,14]. The relationship between vitamin D and glucose homeostasis has been a topic of growing interest in recent years. Low vitamin D levels seem to be associated with insulin resistance disorders and it has been suggested that vitamin D deficiency is one of the factors accelerating the development of insulin resistance [15]. Furthermore, low levels of vitamin D have been implicated in the development of T1D [16] and low levels of 25(OH)D_3_ predict increased risk of all-cause mortality in T1D patients [17].

Although there are several studies linking vitamin D deficiency with T1D incidence and outcomes, there is a lack of information regarding the mechanisms behind this association. In the present study, we generated an inducible double KO mouse (PTEN/VDR) in adulthood, and investigated its effects on survival and glucose/fat metabolism.

## 2. Materials and Methods

### 2.1. Ethical Statement

All animal procedures were approved by the University of Lleida Animal Ethics Committee in accordance with the guidelines of the European Research Council and local laws for the care and use of laboratory animals.

### 2.2. Animal Models

Mice carrying a tamoxifen inducible Cre-estrogen receptor driven by the chicken beta actin promoter/enhancer coupled with the cytomegalovirus immediate-early enhancer (Cre-ERTM) [B6.Cg-Tg(CAG-Cre/Esr1)*5 Amc/J] and floxed homozygous PTEN (C; 129S4-Ptentm1Hwu/J) mice were crossed to generate Cre-inducible PTEN knockout mice (PTEN-KO) as previously described [18]. Mice homozygous for the floxed VDR, generated by the Geert Carmeliet laboratory in a Swiss background [19], were also mated with the Cre-ERTM to generate inducible VDR knockout mice (VDR-KO). Additionally, a double-knockout (DKO) mice was generated by crossing PTEN-KO mice with the mice homozygous for the floxed VDR gene. Cre-negative littermates were used as controls (CNT). Twenty-one days after birth, mice were weaned and genotyped as previously described [20,21]. Primer sequences for genotyping are shown in Appendix A, Appendix A.

To induce Cre-mediated PTEN and VDR ablation, a single intraperitoneal injection of 25 mg/kg tamoxifen solution per 4–5 weeks old mice was given as previously described [18]. Tamoxifen (Sigma-Aldrich) was dissolved in absolute 100% ethanol and then diluted in corn oil (Sigma-Aldrich) to a 5 mg/mL concentration. All the in vivo experiments were performed 4–5 weeks after tamoxifen-induced Cre activation.

Animals were kept in a 12-h light–dark cycle at 22 °C and ad libitum access to water and regular mouse diet (Teklad Global 14% Protein 4% Fat Rodent Maintenance Diet—Envigo. Harlan Teklad, Madison, WI, USA) or high-fat diet (HFD, Paigen Diet 10 MM S9358-E030—1.25% cholesterol, 0.5% cholic acid, 15% cocoa butter, 1% corn oil). Then, 24 h before sacrifice, animals were moved to metabolic cages to quantify food intake. All surgical procedures were performed under general anesthesia with isoflurane. On the day of sacrifice, total body weights were measured. Animals were sacrificed and blood samples were collected by cardiac puncture. Organs were perfused with a saline solution through the left ventricle. Snap-frozen tissue samples were collected for molecular biology and ^3^H glucose analysis and 4% paraformaldehyde-fixed samples for histological assessments.

### 2.3. Glucose Metabolism Analysis

Blood glucose levels were measured with a glucometer (Roche, Basel, Switzerland). To analyze glucose metabolism during fasting, glycemia was analyzed at 2 and 7 h after food removal. Overnight periods of fasting were also used. For the glucose tolerance test (GTT), experiments started 3 h after fasting (time 0) by administration of a glucose (Sigma-Aldrich, St. Louis, MO, USA) single-injection (4 g/kg; i.p.) and glucose measurements were performed at 20, 40, 60 and 120 min. For the pyruvate tolerance test (PTT), the same protocol was followed, replacing glucose with a dose of 2 g/kg sodium pyruvate (Life Technologies, Paisley, UK).

### 2.4. ^3^H Glucose Detection

After 3 h of fasting, a single i.p. bolus of a mixed solution of normal and radioactive glucose (^3^H glucose Perkin Elmer, Westerville, Ohio, USA) 20 μCi per mouse in 3 g/kg glucose) was administered, and animals were sacrificed after 120 min. Then, different tissues were collected and homogenized with stainless steel beads (Qiagen, Hilden, Germany) using a TissueLyser LT (Qiagen). Subsequently, supernatants were mixed 1:1 with 7% HClO_4_ (Sigma-Aldrich), centrifuged and neutralized for 30 min with a 2.2 M KHCO_2_ (Sigma-Aldrich) solution. Then, samples were centrifuged at 14,000× *g*, the precipitate was discarded and supernatant was mixed with 10 mL of scintillation liquid to determine total ^3^H radioactivity [22]. Radioactivity was measured using a scintillation counter (Packard 1900 TR).

### 2.5. Glucose Intraventricular Delivery in the CNS and Osmotic Implantation

Osmotic pumps (Alzet Mini-Osmotic pump model 2006, duration 42 days with a 0.15 μL/h pumping rate) were filled with a saturated solution of D-(+)-glucose or D-mannitol (Sigma-Aldrich) and hydrated with saline at 37 °C for 60 h before surgery. Intracerebroventricular infusion of glucose or mannitol was performed as previously reported by DeVos and Miller [23].

### 2.6. Serum and Urine Biochemistry

Total cholesterol (TC), HDL-C and triglycerides (TG) were measured by colorimetric methods according to standardized protocols with an AU5800 Analyzer (Beckman Coulter Inc., Fullerton, CA, USA) in the Clinical Analysis Laboratory of Arnau de Vilanova University Hospital, in Lleida, Spain. LDL-C was calculated by the Friedewald equation if TG < 250 mg/dL or by a colorimetric method if TG > 250 mg/dL. In order to determine serum c-peptide concentrations, an ELISA kit (Millipore, Bedford, UK) was used as indicated by the manufacturer. The blood urea nitrogen concentration was measured using a colorimetric assay (Spinreact, Barcelona, Spain). Serum 25-hydroxy-vitamin D (25(OH)D_3_) and 1,25-dihydroxy-vitamin D (1,25(OH)_2_D_3_) levels were quantified with an enzyme immunoassay (Immunodiagnostic Systems, Boldon Business Park, UK) following the manufacturer’s instructions.

### 2.7. Histopathology Analysis

Paraffin blocks were cut at 5 µm, dried at 60 °C for 30 min and then dewaxed and rehydrated for hematoxylin (PanReac AppliChem ITW Reagents, Barcelona, Spain) and eosin (Master diagnostic MAD-109 1000) staining. For the periodic acid Schiff-alcian blue (PAS-AB) staining, after rehydration, 5 µm slices were incubated with AB for 5 min, followed by 15 min with PAS, and finally, 25 min in the Schiff solution.

### 2.8. Hepatic Glycogen Detection

Liver samples (100 mg) were homogenized with stainless steel beads (Qiagen) in 500 μL of homogenization buffer (50 mM TrisHCl pH 7.5, 5 mM EDTA, 1 mM DTT, 10 μL/mL PMSF, 5 μL/mL PIC, 5 μL/mL Na_3_VO_4_) using a TissueLyser LT. Then, liver lysates were mixed with 100 μL of 50 mM TrisHCl buffer and 100 μL of 0.2 M perchloric acid and subsequently centrifuged at 14,000× *g* for 5 min. Supernatant was transferred to a new tube with 300 μL of 90% ethanol and maintained at −20 °C overnight. Then, samples were centrifuged and pellets containing the glycogen were let to dry. Once dried, glycogen pellets were mechanically suspended in 2 M HCl. Then, the solution was incubated at 100 °C for 20 min and subsequently the reaction was stopped by adding a 1 M NaOH and 1% 3,5-dinitrosalicylic acid solution. Finally, samples were incubated at 100 °C for 5 min and absorbance was read at 546 nm.

### 2.9. RNA Isolation and Quantitative Reverse Transcription PCR (qRT-PCR)

A total of 20 mg of tissue was used for total RNA isolation from liver samples using an RNA isolation kit (Macherey-Nagel, Allenton, PA, USA) following the manufacturer’s instructions. RNA concentration was measured using a NanoDrop spectrophotometer and stored at −80 °C.

Reverse transcription was performed with 1 µg of total RNA in a reaction buffer (Roche Diagnostics, Mannhein, Germany) containing 5 mM MgCl_2_, 10 µM dNTPs (Biotools, Jupiter, FL, USA), 5 µM random hexamers (Invitrogen, Waltham, MA, USA) and 1 unit of AMV Reverse Transcriptase (Applied Biosystems, Waltham, MA, USA). Then, qPCR reaction using a Taqman Universal PCR Master Mix (Applied Biosystems) was performed as previously described [24]. Primers sequences for qPCR are shown in Appendix A, Appendix A.

### 2.10. Western Blot Analysis

Liver tissue was dispersed into lysis buffer containing 20 mM Tris/HCl pH 7.5, 120 mM NaCl, 0.5% Igepal CA-630 (Sigma-Aldrich), 100 mM NaF, 5 µM PMSF, 5 μL/mL protein inhibitor cocktail (Sigma-Aldrich) and 1 μM Na_3_VO_4_, using a TissueLyser. Protein concentration was determined using a DC protein assay kit (Bio-Rad, Hercules, CA, USA) and Western blot was performed as previously described [24]. Briefly, twenty micrograms of proteins were electrophoresed on 10% SDS-PAGE gels, and transferred to PVDF membranes (Millipore). Membranes were probed with antibodies against PTEN, VDR, AKT and phospho-AKT overnight at 4 °C. Horseradish peroxidase (HRP)-conjugated secondary antibodies (Jackson Immunoresearch, West Grove, PA, USA) and a chemiluminescent kit (EZ ECL, Biological Industries, Kibbutz Beit-Haemek, Israel) or an enhanced chemiluminescent kit (ECL Advanced, Amersham, UK), as appropriate, were used to visualize the amount of each protein. Antibody data are described in Appendix A, Appendix A.

### 2.11. Statistical Analyses

All experiments were carried out at least three times. Statistical analyses were performed with GraphPad Prism 8.02 software. Values are presented as mean ± SEM. Comparisons were assessed using one-way ANOVA followed by Tukey’s test for multiple comparisons with one categorical variable and two-way ANOVA followed by the Sidak test for multiple comparisons with two categorical variables. Survival ratio analyses were performed using the Mantel–Cox test. A *p* < 0.05 was considered to be significant.

## 3. Results

### 3.1. Lack of VDR Reduces Lifespan in PTEN Knockout Mice

To investigate the role of VDR in hypoglycemia induced by overactivation of the insulin receptor, we mated Cre-ERT:PTENfl/fl (PTEN-KO) mice with Cre-ERT:VDRfl/fl (VDR-KO) knockout mice to generate double knockout mice (DKO). At the age of two months, tamoxifen was administered, resulting in genetic excision of the floxed exons after 4–5 weeks (Appendix A) and a low or undetected PTEN protein, and increased AKT phosphorylation in both, PTEN-KO and DKO mice (Appendix A).

All the animals in the CNT group and in the VDR-KO group stayed alive after 65 days of Cre-induced VDR ablation (Figure 1A), whereas animals in the PTEN-KO group showed worse survival (76.2%). Of note, DKO mice presented notable, excessive mortality, starting 20 days after Cre-induced target genes ablation, and resulting in the death of all the animals at 65 days. The patterns of the survival curves in the DKO mice were similar for both sexes (Figure 1B).

### 3.2. Physiological and Biochemical Parameters

The physiological and serum biochemical parameters are shown in Table 1. All animal had ad libitum access to chow, and food intake was increased in PTEN-KO and DKO animals; however, total body weight decreased in both groups as compared with controls.

Serum peptide C concentration was decreased in PTEN-KO and DKO animals as compared with the CNT and VDR-KO groups, indicating decreased insulin secretion, which is associated with lower blood glucose levels. As expected, serum 1,25(OH)_2_D_3_ levels were increased in both the VDR-KO and DKO groups due to the absence of VDR, but decreased in PTEN-KO mice. The serum concentrations of 25(OH)D_3_ were reduced in both PTEN-KO and DKO mice. Total cholesterol (TC) and HDL cholesterol (HDLC) were significantly reduced in PTEN-KO and DKO mice as compared with the VDR-KO group and LDL cholesterol (LDLC) showed a tendency to be reduced in DKO.

### 3.3. Blood Glucose Tests Reveal a Disruption of Glucose Metabolism in PTEN-KO and DKO Mice

Glucose levels were lower in the PTEN-KO and the DKO group, even with unrestricted access to food (Fed state, Figure 2A). Furthermore, after 7 h of food restriction, levels of glucose in the DKO mice had a tendency to be lower than in the PTEN-KO mice. Overnight food restriction led to a 100% mortality in DKO animals.

The results from the glucose and pyruvate tolerance tests are shown in Table 2. Intraperitoneal administration of glucose (Table 2A) produced a significant increase in blood glucose levels in both CNT and VDR-KO mice, which returned to basal values after 2 h. In PTEN-KO animals, the peak was also present, but was smaller and returned to basal values after 60 min. Animals in the DKO group showed a very small increase in glucose levels 20 min after glucose administration, which returned to basal levels at 40 min. Administration of pyruvate (Table 2B) showed similar traits in CNT and VDR-KO mice with glucose peaks that returned to normal values after 2 h. PTEN-KO mice and DKO mice showed a smaller peak in glucose after 20 min, which returned to basal values after 40 min. Therefore, all these results point to a higher utilization of glucose in DKO mice with respect to PTEN-KO mice.

### 3.4. Glucose Supplementation Did Not Increase Survival in DKO Mice

As the DKO group showed severe hypoglycemia, we investigated whether a sucrose supplementation in the drinking water could result in increased survival or extended lifespan. We observed a 100% mortality approximately 100 days after Cre-induced PTEN and VDR ablation, with no differences between the sucrose and vehicle group (Figure 2B).

As severe hypoglycemia leads to functional brain failure and hypoglycemic coma, we also studied whether direct intracerebroventricular infusion of glucose could be beneficial in DKO mice (Figure 2C). We observed that direct infusion of glucose into the cerebral ventricle did not increase animal survival.

We also studied whether glucose consumption was higher in DKO mice that in PTEN mice in a particular organ, as a way to explain the increased utilization of glucose. As shown in Figure 2D, there was a higher intake of glucose in many organs (*p* < 0.01 in the genotype comparison in two-way ANOVA), but the profile of each organ was not modified by the genotype (the interaction was not significant).

### 3.5. Faster Reduction in Glycogen Pool in Fasting DKO Mice

To determine the glycogenic hepatic capacity, total glycogen levels from mouse liver lysates were measured (Figure 3A). In a fed state, all groups showed similar levels of liver glycogen; however, lower glycogen concentration was observed after two hours of fasting in the DKO group as compared with the CNT and VDR-KO groups at the same time point, pointing to a faster use of glycogen stores. However, after 7 h of fasting, all groups showed similar levels of glycogen in liver. Hematoxylin–eosin and PAS staining demonstrated hepatocellular ballooning in the PTEN-KO and DKO mice as compared with the CNT and VDR-KO groups after 7 h of fasting (Figure 3B).

### 3.6. Delayed Gluconeogenesis in PTEN-KO and DKO Mice

In order to study the glucose metabolism in livers, we analyzed the genetic expression of PEPCK and G6PC, genes involved in gluconeogenesis. In CNT and VDR-KO mice, the expressions of PEPCK (Figure 4A) and G6PC (Figure 4B) were increased after 2 h of fasting and were subsequently restored after 7 h of fasting. This fact points to an early induction of glucose generation in the liver that could be already enough to re-establish glycemia. However, in PTEN-KO and DKO PEPCK and G6PC, gene expression reaches a peak after 7 h of fasting, indicating delayed gluconeogenesis.

Additionally, we analyzed the CEBPA and PGC1α gene expression, both genes that are implicated in the transcriptional regulation of PEPCK and G6PC, among other liver metabolism genes. We observed similar liver CEBPA gene expression in the CNT and VDR-KO groups in fed and fasting states, whereas in the PTEN-KO and DKO mice it was downregulated throughout fasting (Figure 4C). PGC1α gene expression also remained similar in CNT and VDR-KO animals in fed and fasting states, however it increased by approximately 3-fold after 7 h of fasting in PTEN and DKO mice, showing earlier upregulation in the DKO group (Figure 4D).

We also measured the gene expression of GLUT2, a glucose transporter found in hepatocytes that mediates glucose diffusion across cell membranes. We observed that GLUT2 gene expression was slightly increased in fasting CNT and VDR-KO mice, whereas it remained similar in the PTEN-KO and DKO groups (Figure 4E).

### 3.7. High-Fat Diet Increased DKO Survival

As fatty acid β-oxidation is used as an alternative source of energy if glucose is not available, we investigated the expression of enzymes related to that process. First, we observed a total absence of abdominal adipose tissue in PTEN-KO and DKO mice (Figure 5A). Second, we investigated several genes involved in fatty acid oxidation. Thus, expression of PPARA was increased after 2 h of fasting, but contrary to the rest of the groups, levels decreased in DKO animals after 7 h of fasting (Figure 5B). Levels of CPT1 were almost unresponsive to starvation in the DKO group, in contrast to the rest of the groups (Figure 5C). Regarding ACOX1 and FGF21, the expression of the genes in the DKO group was attenuated with respect to the PTEN-KO group (Figure 5D,E respectively).

As fat metabolism seems to be involved in the pathological features, we studied the effects of a high-fat diet in the survival rate of DKO mice. Thus, a high-fat diet significantly extended the DKO lifespan (Figure 5F).

## 4. Discussion

In the present study, we generated inducible double KO mice for PTEN and VDR. The animals with deletion of both genes showed an increased mortality, which was partially reverted when animals were placed on a high-fat diet.

PTEN was identified as a tumor suppressor gene on chromosome 10q23 [25]. Vitamin D, signaling through VDR, is also known to have a protective effect against some cancers [26]. Thus, a first possibility is that the elimination of VDR could increase the susceptibility of the animals to develop some kind of cancer. The embryonic lethality of mice with biallelic excision of PTEN has limited the study of complete PTEN ablation in the development of cancer. However, the generation of PTEN conditional-KO mice has solved that problem. Mirantes et al. [18], showed that the use of CREER to delete PTEN-generated mice with a tendency to develop cancer in the thyroid gland, prostate and endometrium. In our case, no macroscopical tumors were detected in the necropsies. Furthermore, the results of the ^3^H glucose uptake showed that the interaction term in the ANOVA analysis was not significant, so a different effect of the phenotype in tissular uptake of ^3^H is ruled out, and an aggravation of the cancer is unlikely to be the cause of death. The differences between our results and those of Mirantes et al. may be explained by two reasons. First, our animals were used 4–5 weeks after the administration of tamoxifen, contrary to the experiments of Mirantes et al. in which they waited 8 weeks for the sacrifice of the animals. Furthermore, to generate our crossings we introduced the SWR/J background into the C57BL6;129S4 mixed background used in the experiments from Mirantes et al., which could change the susceptibility to cancer.

The first important result in PTEN-KO and DKO mice is the lower body weight and the total absence of body fat with respect to the rest of the groups. As animals in these groups had a higher food intake, the results point to an increase in energy expenditure. Previous results have shown conflicting results regarding PTEN and energy expenditure. Thus, the deletion of PTEN in the liver decreases adiposity [27], but systemic overexpression of PTEN has been also shown to increase energy expenditure and decrease adiposity [28]. Constitutive VDR-KO mice also show an increase in energy expenditure and a higher food intake with a lower body weight [29,30]. Our inducible VDR-KO mice also showed a tendency to eat more and have a lower weight. However, those differences did not reach statistical significance with respect to the controls, probably due to the fact that in our case the deletion of VDR was performed in adulthood.

In a previous study by our group, we determined that inducible PTEN-KO mice had alterations in the glucose metabolism that caused hypoglycemia [31]. In addition, vitamin D signaling has been reported to regulate insulin secretion. Thus, vitamin D deficiency inhibits pancreatic secretion of insulin [32] and it is associated with insulin resistance [33]. Furthermore, mice lacking a functional vitamin D receptor show impaired insulin secretory capacity [34] together with insulin resistance [35]. The results of the fasting experiments reported that, although not significant with respect to the PTEN-KO mice, glucose levels had a tendency to be lower in the DKO mice after 7 h of fasting. Thus, it seemed that either glucose was used faster or produced in a lower rate in those mice. The glucose tolerance test results showed that although a smaller peak of glucose could be seen 20 min after glucose administration in the DKO mice with respect to the PTEN-KO mice, the levels dropped fast, achieving similar values in both groups after 40 min. Therefore, a significant effect due to a higher uptake of glucose in the tissues of DKO mice seems unlikely, as it was also shown in the ^3^H-glucose experiments. The PTT also showed no significant differences in the rate of gluconeogenesis between both phenotypes. Furthermore, and although DKO animals seemed to decrease glycogen storage at a faster rate after 2 h of fasting, the differences were not significant after 7 h. Interestingly, after an overnight fasting period, 100% of the DKO animals died. When glucose levels in blood achieve a lower threshold, symptoms and signs of encephalopathy result. The blood glucose level at which cerebral metabolism fails and symptoms develop varies, but in general, confusion occurs at levels below 30 mg/dL and coma below 10 mg/dL [36]. Thus, it is possible that hypoglycemic coma in our animals, induced by an overactivation of the insulin receptor similar to an insulin overdose, can cause unresponsiveness and inability to feed and drink, causing the death of the animals. However, supplementation with sucrose in the drinking water was also unable to decrease the mortality observed in the DKO group. Furthermore, the experiments in which glucose was infused directly into the cerebral ventricles with an osmotic pump were also unable to reduce the excess mortality in the DKO mice.

The mechanisms by which insulin-induced hypoglycemia causes sudden death are not well characterized. It has been previously shown that fatal arrhythmias and seizures are involved in this fatal complication [37]. The influence of high cholesterol levels on cardiovascular mortality has been known for decades. However, recent results point to a deleterious effect of low lipid levels on cardiovascular events, especially in atrial fibrillation (AF). Thus, a study on 15 million Chinese participants showed that low HDLC was independently associated with a higher risk of atrioventricular block, whereas high TC was a protective factor [38]. The protective effect of high levels of TC against AF has been demonstrated in many other studies in different populations in Korea, Japan, USA, China and Sweden [39,40,41,42,43]. Low HDLC has been also found to be associated with increased risk of AF [39,44,45,46]. In our mice, lower levels of TC and HDLC were found in PTEN-KO mice after 7 h of fasting, and showed a tendency to be even lower in DKO mice. Furthermore, the maintenance of the animals on a high-fat diet was the only strategy able to increase its lifespan.

In fasting adult mammals, 60–80% of cardiac energy metabolism relies on the oxidation of fatty acids (FAs) with glucose, lactate, and ketones providing substrates for the remainder [47]. Both PTEN and vitamin D show effects on hepatic lipid metabolism. Thus, liver specific PTEN-KO mice show increased fatty acid synthesis, accompanied by hepatomegaly and fatty liver phenotype [48]. In contrast, VDR deletion induces lipid oxidation and fat consumption in hepatocytes [12]. In our inducible PTEN-KO mice, we observed that fasting-mediated induction of PPARA, a regulator of hepatic metabolism activated by fatty acids [49], was not as elevated as in CNT and VDR-KO mice. This induction was further reduced in the DKO mice, reaching basal levels after 7 h of fasting. PPARA controls gene expression levels of the rate-limiting enzymes of peroxisomal β-oxidation, including ACOX1 [49], which showed a similar profile to PPARA. Another gene implicated in fatty acid metabolism is CPT1. The protein encoded by CPT1 is responsible for the carnitine-dependent transport of fatty acids across the mitochondrial inner membrane. In our model, CPT1 expression was increased by starvation in all the groups except in the DKO animal. Finally, another PPARA-mediated target, FGF21, was increased in PTEN-KO animals but it did not show the same profile in DKO animals. The effects of FGF21 on the liver are not completely understood but reports show that it stimulates the oxidation of fatty acids [50]. Therefore, it seems that in the DKO mice, together with the alterations in glucose metabolism leading to hypoglycemia, alterations in lipid metabolism leading to delays in the fatty acid oxidation pathways can increase the mortality rate.

## 5. Conclusions

Taken together, the results shown in the present paper point to the paramount role of an adequate vitamin D signaling pathway in hypoglycemia induced by overactivation of the insulin receptor. Thus, in T1 diabetic patients, especially in the lean phenotype, maintaining correct levels of vitamin D could support proper lipid metabolism and decrease deaths induced by insulin dosing errors.

## Figures and Tables

**Figure 1 nutrients-14-01516-f001:**
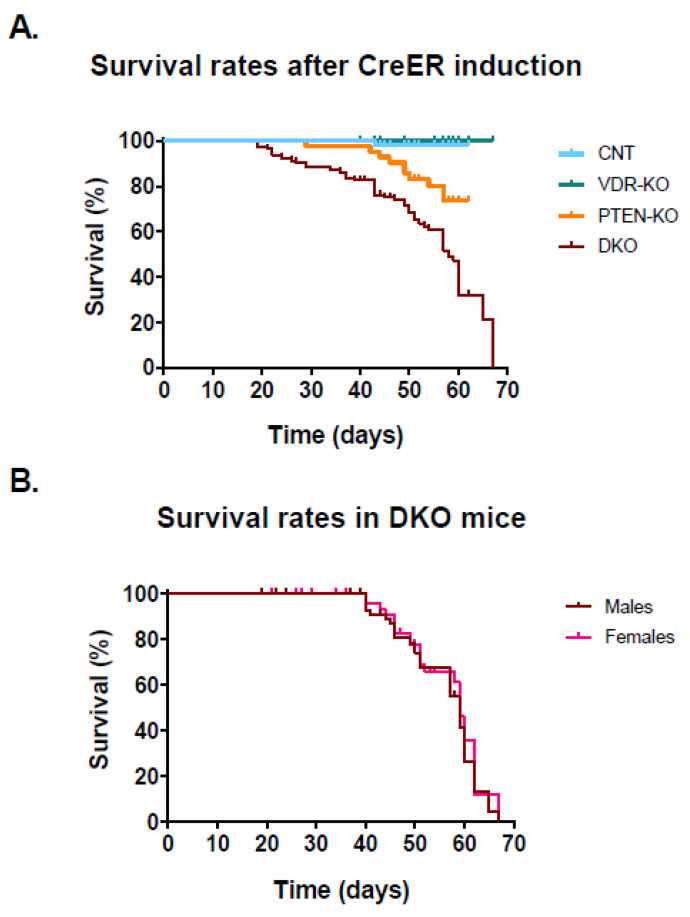
Survival rates after Cre-induced PTEN ablation. (**A**) The 65-day follow up after tamoxifen injection showed a 98.2% survival rate in CNT (54 of 55), 100% in VDR-KO (42 of 42), 76.2% in PTEN-KO (32 of 42) and 0% in DKO (0 of 59); long-rank test *p* < 0.001. (**B**) Survival rate was not influenced by sex; Long-Rank test *p* = 0.472. CNT: Control. VDR-KO: Vitamin D receptor knockout. PTEN-KO: Phosphatase and tensin homolog knockout. DKO: Double knockout.

**Figure 2 nutrients-14-01516-f002:**
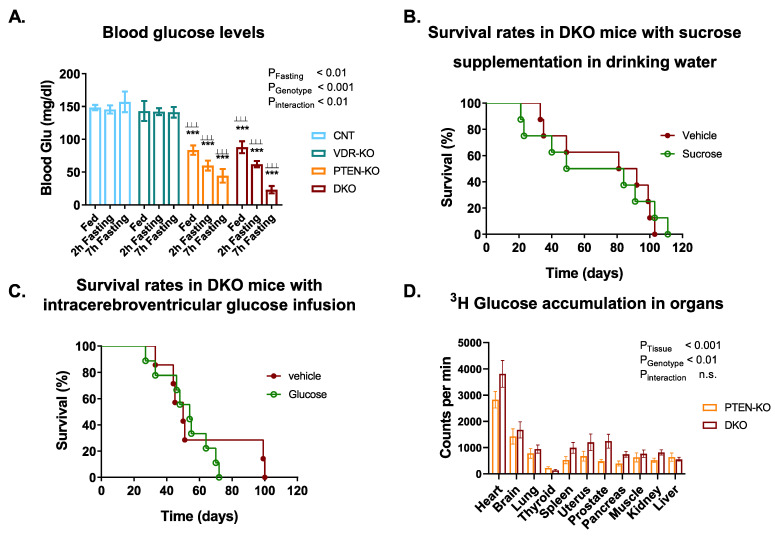
Administration of glucose did not increase survival in DKO. (**A**) Blood glucose levels in fed and fasting states. (**B**) Sucrose supplementation in drinking water or (**C**) intracerebroventricular infusion of glucose did not extend DKO survival, Long-Rank test *p* = 0.91 and 0.57 respectively. (**D**) Tissue glucose accumulation in PTEN-KO and DKO mice. ^3^H-Glucose radioactivity was measured in 10 mg of different tissues. Data are represented as mean ± SEM. *** *p* < 0.001 vs. CNT; ^⊥⊥⊥^
*p* < 0.001 vs. VDR-KO. CNT: Control. VDR-KO: Vitamin D receptor knockout. PTEN-KO: Phosphatase and tensin homolog knockout. DKO: Double knockout.

**Figure 3 nutrients-14-01516-f003:**
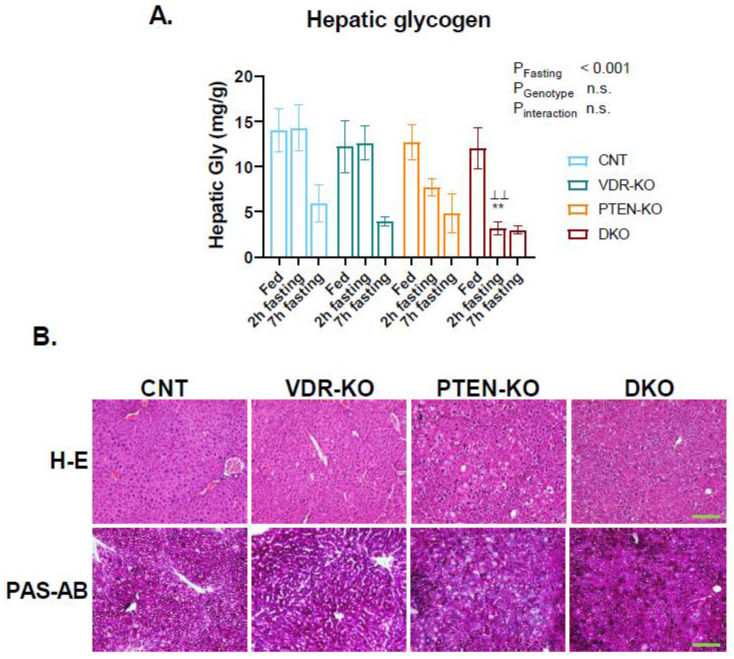
Abnormal glycogen metabolism in PTEN-KO and DKO mice. (**A**) Hepatic glycogen in the fed state and after 2 h and 7 h of fasting. (**B**) Representative microphotographs of liver hematoxylin-eosin and PAS staining for the groups of study. Data are represented as mean ± SEM. ** *p* < 0.01 vs. CNT; ^⊥⊥^
*p* < 0.01 vs. VDR-KO. 100× magnification. Scale bar: 100 µm. CNT: Control. VDR-KO: Vitamin D receptor knockout. PTEN-KO: Phosphatase and tensin homolog knockout. DKO: Double knockout.

**Figure 4 nutrients-14-01516-f004:**
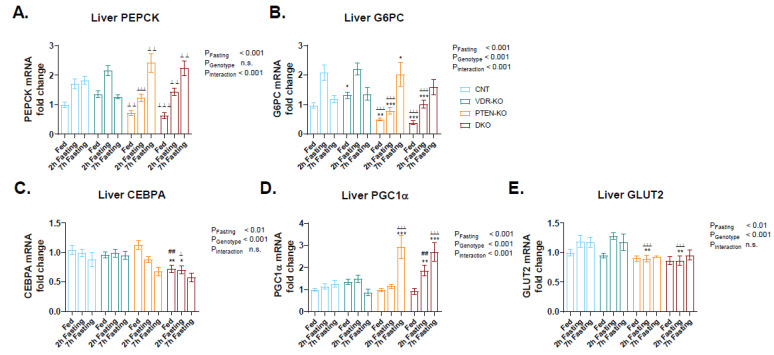
Expression of genes involved in glycogen metabolism in liver. mRNA expression of (**A**) Phosphoenolpyruvate carboxykinase (PEPCK), (**B**) Glucose-6-phosphatase (G6PC), (**C**) CCAAT/enhancer-binding protein alpha (CEBPA), (**D**) Peroxisome proliferator-activated receptor gamma coactivator 1-alpha (PGC1α) and (**E**) Glucose transporter 2 (GLUT2) using TATA-binding protein (TBP) as the housekeeping gene. Data are represented as mean ± SEM. * *p* < 0.05 vs. CNT; ** *p* < 0.01 vs. CNT; *** *p* < 0.001 vs. CNT; ^⊥^
*p* < 0.05 vs. VDR-KO; ^⊥⊥^
*p* < 0.01 vs. VDR-KO; VDR-KO; ^⊥⊥⊥^
*p* < 0.001 vs. VDR-KO; ## *p* < 0.01 vs. PTEN-KO. CNT: Control. VDR-KO: Vitamin D receptor knockout. PTEN-KO: Phosphatase and tensin homolog knockout. DKO: Double knockout.

**Figure 5 nutrients-14-01516-f005:**
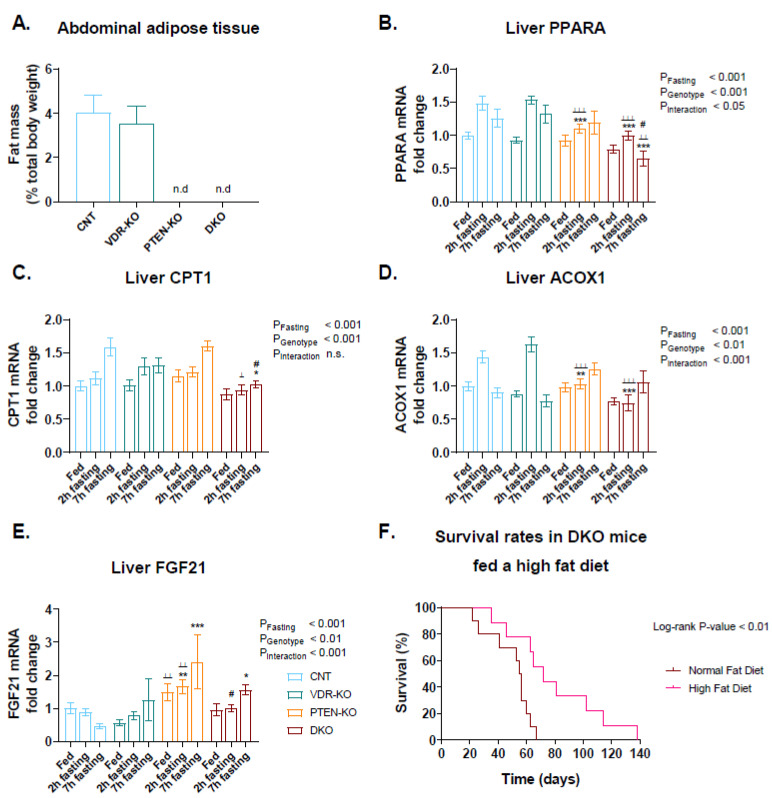
**A** high-fat diet extends lifespan in DKO mice. (**A**) Abdominal adipose tissue was absent in PTEN-KO and DKO mice. Liver mRNA expression of (**B**) Peroxisome proliferator-activated receptor alpha (PPARA), (**C**) Carnitine palmitoyltransferase I (CPT1), (**D**) Peroxisomal acyl-coenzyme A oxidase 1 (ACOX1) and (**E**) Fibroblast growth factor 21 (FGF21) using TATA-binding protein (TBP) as housekeeping gene. (**F**) Survival rates of DKO mice fed on a normal or high fat diet after cre-induced PTEN ablation. Log-rank *p*-value < 0.01. Data are represented as mean ± SEM. n.d: not detected. * *p* < 0.05 vs. CNT; ** *p* < 0.01 vs. CNT; *** *p* < 0.001 vs. CNT; ^⊥^
*p* < 0.05 vs. VDR-KO; ^⊥⊥^
*p* < 0.01 vs. VDR-KO; VDR-KO; ^⊥⊥⊥^
*p* < 0.001 vs. VDR-KO; # *p* < 0.05 vs. PTEN-KO. CNT: Control. VDR-KO: Vitamin D receptor knockout. PTEN-KO: Phosphatase and tensin homolog knockout. DKO: Double knockout.

**Table 1 nutrients-14-01516-t001:** Physiological and serum parameters.

	CNT (*n* = 11)	VDR-KO (*n* = 9)	PTEN-KO (*n* = 8)	DKO (*n* = 13)
A. Physiological parameters				
Food intake (g/24 h)	3.03 ± 0.28	3.87 ± 0.29	5.02 ± 0.35 **	5.26 ± 0.31 ***
Total body weight (g)	27.0 ± 0.81	25.0 ± 1.01	22.7 ± 0.50 ***	22.6 ± 0.58 ***
B. Circulating parameters				
BUN (mg/24 h)	20.6 ± 2.74	19.4 ± 1.83	20.4 ± 1.45	20.6 ± 2.41
Peptide-C (pM)	175.7 ± 25.3	140.9 ± 16.1	64.0 ± 3.61 *** ^⊥⊥^	70.0 ± 6.50 *** ^⊥⊥^
Blood glucose (mg/dL)	148.4 ± 4.11	143.1 ± 15.2	83.5 ± 7.12 *** ^⊥⊥⊥^	87.9 ± 9.03 *** ^⊥⊥⊥^
25(OH)D_3_ (ng/mL)	74.7 ± 10.2	65.04 ± 18.9	17.8 ± 3.15 *** ^⊥⊥^	26.1 ± 7.05 **
1,25(OH)_2_D_3_ (pmol/L)	124.7 ± 39.7	376.3 ± 67.5 ***	70.10 ± 40.4 ^⊥⊥⊥^	188.0 ± 41.5 ^##^
Total cholesterol (mg/dL)	121.3 ± 6.05	142.0 ± 13.0	110.1 ± 6.44 ^⊥^	101.3 ± 3.85 ^⊥⊥^
LDL cholesterol (mg/dL)	16.9 ± 2.43	16.7 ± 3.77	16.0 ± 4.45	9.23 ± 2.50
HDL cholesterol (mg/dL)	92.0 ± 4.86	111.4 ± 9.70	83.71 ± 5.55 ^⊥^	79.4 ± 3.52 ^⊥⊥^
Triglycerides (mg/dL)	88.3 ± 12.7	69.4 ± 5.63	93.3 ± 9.11	64.7 ± 8.41

Values represent mean ± SEM. ** *p* < 0.01 vs. CNT; *** *p* < 0.001 vs. CNT; ^⊥^
*p* < 0.05 vs. VDR-KO; ^⊥⊥^
*p* < 0.01 vs. VDR-KO; ^⊥⊥⊥^
*p* < 0.001 vs. VDR-KO; ^##^
*p* < 0.01 vs. PTEN-KO. BUN: Blood urea nitrogen. 25(OH)D3: 25-hydroxy-vitamin D. 1,25(OH)2D3: 1,25-dihydroxy-vitamin D. LDL: Low-density lipoprotein. HDL: High-density lipoprotein. CNT: Control. VDR-KO: Vitamin D receptor knockout. PTEN-KO: Phosphatase and tensin homolog knockout. DKO: Double knockout.

**Table 2 nutrients-14-01516-t002:** Glucose and pyruvate tests.

	CNT (*n* = 11)	VDR-KO (*n* = 9)	PTEN-KO (*n* = 8)	DKO (*n* = 13)
A. Glucose tolerance test				
Time (min)				
0	145.4 ± 6.3	134.0 ± 5.3	61.5 ± 6.7 *** ^⊥ ⊥ ⊥^	68.0 ± 4.7 *** ^⊥ ⊥ ⊥^
20	190.1 ± 7.3	202.2 ± 8.8	96.7 ± 17.0 *** ^⊥ ⊥ ⊥^	72,7 ± 6.9 *** ^⊥ ⊥ ⊥^
40	180.7 ± 7.7	197.2 ± 11.8	68.2 ± 9.0 *** ^⊥ ⊥ ⊥^	64.8 ± 5.8 *** ^⊥ ⊥ ⊥^
60	175.0 ± 9.0	195.9 ± 11.9	58.8 ± 8.6 *** ^⊥ ⊥ ⊥^	65.3 ± 6.7 *** ^⊥ ⊥ ⊥^
120	148.3 ± 7.5	155.5 ± 16.9	54.8 ± 7.0 *** ^⊥ ⊥ ⊥^	51.2 ± 6.2 *** ^⊥ ⊥ ⊥^
B. Pyruvate tolerance test				
Time (min)				
0	157.1 ± 7.3	149.1 ± 7.2	62.1 ± 10.5 *** ^⊥ ⊥ ⊥^	68.4 ± 9.2 *** ^⊥ ⊥ ⊥^
20	212.3 ± 8.9	196.2 ± 14.6	74.3 ± 11.2 *** ^⊥ ⊥ ⊥^	78.6 ± 9.9 *** ^⊥ ⊥ ⊥^
40	204.6 ± 17.2	175.0 ± 20.6	62.9 ± 11.7 *** ^⊥ ⊥ ⊥^	54.1 ± 6.6 *** ^⊥ ⊥ ⊥^
60	204.1 ± 18.4	180.8 ± 20.0	61.5 ± 14.0 *** ^⊥ ⊥ ⊥^	49.8 ± 6.7 *** ^⊥ ⊥ ⊥^
120	145.3 ± 10.7	148.9 ± 19.4	59.8 ± 7.9 *** ^⊥ ⊥ ⊥^	60.3 ± 12.9 *** ^⊥ ⊥ ⊥^

Time 0 is 3 h after food removal. Values represent mean ± SEM. *** *p* < 0.001 vs. CNT; ^⊥ ⊥ ⊥^
*p* < 0.001 vs. VDR-KO. CNT: Control. VDR-KO: Vitamin D receptor knockout. PTEN-KO: Phosphatase and tensin homolog knockout. DKO: Double knockout.

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
