# Peer review of "Elimination of Vitamin D Signaling Causes Increased Mortality in a Model of Overactivation of the Insulin Receptor: Role of Lipid Metabolism"

_nutrients, 2022, doi:10.3390/nu14071516_

Round 1

Reviewer 1 Report

This experimental animal study uses inducible double KO (DKO) mice of the VD receptor (VDR) and PTEN.
DKO mice showed severe hypoglycemia, lower total cholesterol, and increased mortality. No macroscopic tumors were detected. Analysis of the
 glucose metabolism did not show apparent differences that would explain the increased mortality. Glucose supplementation, either systemically or 
directly into the brain, did not enhance DKO survival. Lipidic liver metabolism was altered as there was a delay in activating genes related to 
oxidation and a decrease in lipogenesis in DKO mice. High-fat diet administration in DKO significantly improved its life span.

The study concept is innovative, and the study design is complete.
The analysis of the result data is appropriate.

However, the clinical implication of these animal experiments needs more addressing.
The DKO mice seem to lack the ability to use blood glucose as an energy source.
The High-Fat-Diet mice have better survival because they receive adequate energy from the lipid.
The author should discuss the impact of this high lipid-based diet on the IR deficiency model for survival and 
the conclusion regarding type I DM patients.  

Author Response

We thank the reviewer for the analysis of the manuscript. We are a little reluctant to discuss further the clinical implications of the findings, as the results can should be considered just preliminary as they are produced in a very aggressive model. Thus KO models are very interesting as they can uncover hidden relationships, but are seldomly found in the clinic as a total lack of signaling is difficult to find.

Reviewer 2 Report

This manuscript is an initial exploration of the role of the VDR in modulating insulin sensitivity in animals. As such, it is difficult to draw conclusions about vitamin D that would be useful for humans, but the authors do a good job of limiting their discussion to the results provided.

There are no major with this manuscript. One question that might merit answering in the paper: Are there any connections between a VDR knockout and reduced insulin sensitivity? It is noted that the glucose levels in the VDR-KO animals remain high in the tolerance test, suggesting poor(er) insulin signaling in those animals. 

Author Response

We thank the reviewer for the comments. We have now included a few lines in the discussion section regarding the association between VDR and insulin resistance.